# Spectral Diffractive Lenses for Measuring a Modified Red Edge Simple Ratio Index and a Water Band Index

**DOI:** 10.3390/s21227694

**Published:** 2021-11-19

**Authors:** Veronika Blank, Roman Skidanov, Leonid Doskolovich, Nikolay Kazanskiy

**Affiliations:** 1Image Processing Systems Institute of RAS—Branch of the Federal Scientific Research Centre Crystallography and Photonics of Russian Academy of Sciences, 151 Molodogvardeyskaya St., 443001 Samara, Russia; veronika_b@ipsiras.ru (V.B.); leonid@ipsiras.ru (L.D.); kazanskiy@ssau.ru (N.K.); 2Department of Technical Cybernetics, Samara National Research University, 34 Moskovskoe Shosse, 443086 Samara, Russia

**Keywords:** diffractive lenses, spectral lenses, vegetation indices, hyperspectrometer

## Abstract

We propose a novel type of spectral diffractive lenses that operate in the ±1-st diffraction orders. Such spectral lenses generate a sharp image of the wavelengths of interest in the +1-st and –1-st diffraction orders. The spectral lenses are convenient to use for obtaining remotely sensed vegetation index images instead of full-fledged hyperspectral images. We discuss the design and fabrication of spectral diffractive lenses for measuring vegetation indices, which include a Modified Red Edge Simple Ratio Index and a Water Band Index. We report synthesizing diffractive lenses with a microrelief thickness of 4 µm using the direct laser writing in a photoresist. The use of the fabricated spectral lenses in a prototype scheme of an imaging sensor for index measurements is discussed. Distributions of the aforesaid spectral indices are obtained by the linear scanning of vegetation specimens. Using a linear scanning of vegetation samples, distributions of the above-said water band index were experimentally measured.

## 1. Introduction

Hyperspectral remote sensing vegetation imagery has found uses in environmental monitoring, agriculture, forestry, urban green infrastructures, and so on. In particular, applications of hyperspectral imagery in agriculture include the assessment of the current condition of fields, crops yield, and detection of crop diseases [1,2,3,4]. In recently published works, it was proposed that imaging hyperspectrometers be directly mounted on the farming machinery, thus enabling some target vegetation parameters, such as the transient crop moisture content, to be estimated in a possibly quick manner [5,6]. A characteristic feature of the vegetation and its current condition is described by the spectral reflectance, which is widely varying for different wavelengths. However, the most common approach to assessing target vegetation parameters is based on the analysis of two to four wavelengths, which utilizes the so-called spectral vegetation indices [7,8,9,10,11,12,13,14,15,16,17]. The vegetation indices represent algebraic relationships of a combination of the reflectance of the object under analysis for several narrow spectral bands, which are indicative of values of the target parameters of the object under study (e.g., the moisture content in the vegetation cover). Thus, out of a large number of spectral channels of the imaging hyperspectrometer (50–300 channels), just 5 percent are really utilized. Considering that a full-range hyperspectrometer is rather costly, it would be inexpedient to mount it on an irrigating machine [5], which uses a single vegetation index. A possible way to address this problem is through simplifying the hyperspectrometer design, with many recent publications discussing issues of the miniaturization and reduction of cost of the hyperspectrometers and multispectrometers [18,19,20,21,22,23,24,25,26,27,28,29]. Notably, some papers offered very unconventional approaches. For instance, Ref. [29] discussed the use of a special narrowband illumination source in place of a sophisticated setup for wavelength separation, which has made it possible to create a very cheap setup for multispectral imaging, though applicable only in laboratory conditions. However, it seems appropriate to design a device specifically intended for vegetation index measurements. Works that rely on such an approach have already been published, with a diffractive lens array utilized for wavelength separation in Ref. [30], resulting in a compact setup for measuring the NDVI index but suffering from high aberrations in the off-axis region. In this work, we propose the design of a spectral device for measuring a single vegetation index based on spectral diffractive lenses (SDL). A method for designing the SDLs was described in Refs. [31,32]. The SDLs enable several wavelengths of light to be focused in the same plane but in different diffraction orders, e.g., with wavelengths λ_1_ and λ_2_ being, respectively, imaged in the +1-st and –1-st diffraction orders. In fact, as a combination of the microreliefs of a harmonic lens and a beam-splitting diffraction grating, the designed scheme is the development of an idea proposed in Ref. [24].

Spectral diffractive lenses, which focus light of specified wavelengths corresponding to one or several spectral indices at different points, can be utilized when designing simple and compact sensors for the real-time monitoring of the vegetation cover and in specialized agricultural machinery [5,6]. In this work, we discuss the SDLs for measuring the following VIs: a modified red edge simple ratio index and water band index. The former enables detecting the presence of vegetation while the latter measures its relative moisture. The water band index is of special importance for real-time monitoring because sometimes it can essentially vary in the course of minutes.

## 2. Synthesis of Spectral Diffractive Lenses

The method for designing spectral lenses has been described in detail [31] and so we do not dwell on it at length in this work.

For the experimental study, we have chosen SDLs for measuring vegetation indices mSR705 and WBI, i.e., a modified red edge simple ratio index at wavelengths of 455 nm and 750 nm and a water band index at wavelengths of 900 nm and 970 nm. The SDLs were synthesized by direct laser writing on a laser writing station CLWS-2014 in a 6-µm-thick photoresist FP-3535 preliminarily applied on a silicon substrate by centrifuging. The resulting microrelief was of ~4-µm thickness.

Lenses of diameter 4 mm and focal length f = 70 mm were used. Illustrations in Figure 1a and Figure 2a; depict optical microscope images of SDL microreliefs for separating the wavelengths 455 nm and 750 nm, and 900 nm and 970 nm, respectively.

Figure 1b and Figure 2b depict microrelief fragments with a clearly seen fine structure of the SDLs, while Figure 1c and Figure 2c depict profilograms measured relative to the lens symmetry axis using a Tencor profiler.

## 3. Experiments with a Tunable Laser

A tunable laser was used to accurately measure a point spread function of the SDLs at the wavelengths under study. The experimental optical scheme used is presented in Figure 3.

The tunable laser NT-242 generates a laser beam of a specified wavelength. Lens 2 focuses the beam onto a 10-µm pinhole before Lens 4 produces a collimated beam with a divergence angle of 0.0001°. The collimated beam then falls on the SDL, which focuses light onto a photosensitive matrix of the Basler acA 1920-40um camera (Basler, Ahrensburg, Germany). Figure 4a,b, respectively, show intensity patterns for wavelengths of 455 nm and 750 nm (vertical bars mark positions of the diffraction orders –1, 0, and +1). The zero diffraction order is found exactly at the image center. Figure 4c,d show magnified images of the point spread function for the wavelengths 455 nm and 750 nm, respectively.

From Figure 4, the light is really seen to be focused at the diffraction orders +1 and –1, with the PSF being ~10 µm for both wavelengths.

Figure 5a,b, respectively, depict intensity patterns for the wavelength of 900 nm and 970 nm, with the vertical bars marking positions of the orders –1, 0, and +1. The zero diffraction order is located exactly at the image center. Figure 4c,d show magnified images of the PSF for wavelengths of 900 nm and 970 nm.

From Figure 4 and Figure 5, the PSF width at half-height is seen to be less than 10 µm, showing that the image is of high quality and there is a potential feasibility for developing SDL-based sensors. Such sensors may find uses for measuring the hyperspectral VI of interest with high spatial resolution.

## 4. Experiments with a Broadband Source

The next series of experiments aimed to simulate the operation of an SDL as a sensor component for vegetation index measurements. In initial experiments, a 0.4–0.5-mm pinhole in an opaque screen put in front of a high-power halogen lamp served as a small light source (Figure 6). The light source and a lens were at a distance of ~1 m. Besides, to simulate the operation of an imaging hyperspectrometer, in which objects are commonly scanned through a slit diaphragm, we experimented with several 0.4–0.5-mm-long light sources approximately arranged along a line.

Thanks to a comparatively small distance to the object, the SDL-aided image was generated at a working distance of 75 mm, where a photosensitive matrix of the Basler acA 1920-40umcamera (Basler, Ahrensburg, Germany) was located. The experimental results are shown in Figure 7. In Figure 7a, the photosensitive matrix is seen to have registered two diffraction orders, namely, –1 for a wavelength of 455 nm and +1 for 750 nm, with the line-arranged group of sources imaged as two lines at diffraction orders –1 and +1 for the above-said wavelengths.

The diffraction order at 750 nm is seen to be brighter thanks to an essential difference in the spectral intensities (Figure 8).

From Figure 8, the lamp spectral intensity for 750 nm is seen to be seven times that for 455 nm, which accounts for the intensity difference in the diffraction orders in Figure 7.

The SDL is seen to form sharp images of non-point sources in the +1st and –1st diffraction orders, opening up a possibility for SDL-aided imaging both in a full-frame regime and a scanning-through-slit regime. In other words, it becomes possible to image fairly wide swaths using a 100-um slit, or wider. With this approach, the process of obtaining VI imagery can be essentially accelerated.

An experiment was also conducted with an SDL intended for measuring wavelengths of 900 nm and 970 nm for calculating a hyperspectral water band index. With these wavelengths being much closer to each other compared to the previous experiment, they will be imaged too close on the photosensitive matrix when using a hyperspectrometer with a conventional diffraction grating. With the proposed SDL, the two wavelengths are separated at sufficiently distant different diffraction orders (Figure 9).

From Figure 9, the orders are seen to have the near-same brightness, which is due to the near-same spectral intensity of the light source at the wavelengths used. Figure 10 illustrates the possibility of the instantaneous imaging of fairly wide swaths instead of scanning through a narrow slit. To calculate the area covered when using slit-aided scanning, we used an ISO test pattern 12233:2000 (Figure 10a) used for measuring the electron camera resolution. Figure 10b shows a fragment of the test pattern image obtained at 900 nm and 970 nm in the –1st and +1st diffraction orders, respectively.

From Figure 10b, images of fairly good quality are seen to be formed in both diffraction orders. Meanwhile, thanks to a considerably large coverage area, the object scanning process can be essentially accelerated when compared with conventional hyperspectral imaging where the slit width commonly needs to be matched with the photosensitive camera pixel size.

In the SDLs synthesized, both optical power and diffraction efficiency are relatively low [31], so it would be technically challenging to use them ‘as is’ in the field experiments.

So, the experiment on obtaining vegetation index images was conducted in the laboratory. The experimental optical setup in Figure 11 placed in a light-proof case (which is removed in the illustration) comprised an objective lens 1, a slit diaphragm 2, a spectral diffractive lens SDL 3, and a Basler acA 1920-40 um camera 4 (Basler, Ahrensburg, Germany).

In the experiment, a LED-projector source illuminated a vegetable object placed on a green polycarbonate substrate to complicate the experimental conditions. The object was put on a linearly moving table to perform a scanning process. Using objective 1, the object was imaged in the plane of slit diaphragm 2 whose width was varied from 80 um to 800 um, before SDL 3 generated a spectral image on a photosensitive matrix of the Basler acA 1920-40um4 camera (Basler, Ahrensburg, Germany), with images for other wavelengths later being formed similarly.

Objects for VI measurements included tree leaves, namely, those of elm and linden, which are easily available in an urban area. The leaves were glued to the substrate, as shown in Figure 12. The upper part was exposed to the high-power heat from a 1500 W halogen lamp from a 1-m distance for 16 min, thus making the exposed parts essentially drier. The duration and mode of the drying procedure were fitted experimentally. The aim was to achieve a tint of the dried-up leaf parts visually indiscernible from that of the wet leaf parts to be able to check the effectiveness of the use of the water band index because otherwise, the leaf dryness would be easier to estimate using an RGB image. As a result of the drying procedure, the upper parts of the leaves got essentially drier while the lower parts were shielded with a mirror. The leaf humidity was checked using conductometry, via measuring the electric resistance of the leaves. The wet leaf parts were found to have a resistance of 200–280 kOhm, while the dried-up leaf parts had an essentially higher resistance of 11–30 MOhm, indicating an essentially lower water content in the latter.

In the course of the experiments, the objects were illuminated with a high-power 150-W LED projector which provided illumination levels similar to those from the midday sunshine in terms of power and energy distribution, while almost not heating the leaves to avoid fast-drying of their other parts. For obtaining a hyperspectral image, the objects under study were moved using a motorized moving table. Several dozens of experiments that were conducted aimed to obtain water band index images that would be indicative of the nonuniform moisture content in the leaves. Figure 13 depicts (a,b) SDL-aided images of elm leaves, respectively, acquired at 900 nm and 970 nm, and (c) a water band index image for the elm leaves from Figure 12a and a scale of index values.

A similar experiment was conducted for linden leaves. Unlike the experiments with the elm leaves, the linden leaves were dried over almost their entire area, except for tips (bottom) but the 12-min drying procedure was milder. In doing so, the purpose was to look into the possibility of using a water band index with a higher-than-two number of gradations. While in the experiments with the elm leaves there were only two possible outcomes—dry fragment vs. wet fragment—in the experiment with the linden leaves the moisture content was to vary over a fairly wide range due to nonuniform drying of the leaf surface. Measurements of the electric resistance showed that it changed from 270–320 kOhm to 1.0–2.0 MOhm, strongly depending on where the ohmmeter probe got in touch with the leaf surface.

Figure 14 depicts (a,b) SDL-aided images of linden leaves, respectively, acquired at 900 nm and 970 nm, and (c) a water band index image for the linden leaves from Figure 12b and a scale of index values.

In Figure 13c and Figure 14c, most wet leaf parts are the brightest and the dried up leaf parts are the darkest.

Thus, the results obtained for the index image are as one would expect. The more dried-up fragment at the top of the image contains both light and gray, and even black fragments, corresponding to fully wet and fully dried-up leaf parts. Unfortunately, using conductometry it is not possible to determine the moisture of small leaf fragments in such detail. To look into the potentiality of the use of the water band index with a large number of identifiable moisture levels, one needs either to employ pointwise measurement techniques of leaf moisture or to conduct in future field experiments where it would be possible to measure the moisture of individual plants.

## 5. Discussion

The SDLs proposed herein are a combination of diffractive lenses and beam-splitting gratings. Both types of diffractive structures have been known and utilized for a long time. However, their combined use in a single microrelief opens up opportunities for brand new applications in imaging spectroscopy. Actually, SDL-based imaging systems can be used for obtaining index images. Besides, a considerable distance between separate orders enables the imaging to be performed using a technique essentially different from the narrow slit-aided scanning. Normally, in a scanning imaging spectrometer, the slit width is matched with the pixel size of a photosensitive matrix, so that the slit image on the matrix equals the matrix pixel. Meanwhile, in the SDL-based design proposed herein, the slit width depends on the inter-order distance, with the main condition imposed on the slit width being that the diffraction orders do not superpose. In our experiments, the diffraction orders were located at a distance of 0.35 mm for the Modified Red Edge Simple Ratio Index and 0.5 mm for the Water Band Index, making it possible in the latter case, to bring the slit width to 0.8 mm, considering that the distance of the SDL to the slit was 170 mm, i.e., 2.5-times the SDL focal length. We would like to note that in this work it was not our purpose to determine precise values of leaf moisture; we just aimed to demonstrate that using the water band index it is possible to qualitatively distinguish between wet and dry leaves while there is no clear visual difference. An attempt to obtain a larger number of water band index gradations in the experiment with linden leaves turned out to be only partially successful due to the lack of a precise method of moisture measurements in small leaf fragments. It will only be possible to investigate the accuracy of leaf moisture classification at the following stages of the study after spectral lenses suitable for field experiments will have been synthesized.

Parameters of diffractive spectral lenses designed and fabricated as part of this work were fitted to make the fabrication process simpler, and before they can be used in field experiments, the DSL parameters will need to be modified. For instance, the DSL diameter will need to be increased while reducing its focal length, in this way increasing the system aperture ratio and reducing its size. However, this will lead to a more complicated fabrication procedure because this means an essentially smaller spectral lens period on the periphery.

The proposed optical elements have been analyzed just for two wavelengths but the number of wavelengths such a lens can focus on is unlimited. It is potentially possible to synthesize sensors for measuring three-to-four wave indices. We also note that in terms of practical uses, it is important to increase the number of moisture content gradations detectable (potentially bringing them to 10–12). We envisage that this can be achieved in later field experiments using DSL with a higher aperture ratio, which will enable the moisture content of individual plants in the image to be measured using conventional accurate measurement techniques.

## 6. Conclusions

Based on the study results, we can infer that SDLs are well suited for acquiring vegetation index images. Besides, thanks to the essential spatial separation of different spectral components, the use of SDL-based schemes makes it possible to form index images in an interim regime between the full-frame imaging using narrow-band filters and scanning with a narrow slit diaphragm. In this work, we implemented an instantaneous imaging regime for fairly wide swaths, followed by sewing a full-frame index image.

A water band index image used to detect the vegetation moisture content has been experimentally obtained and shown to be in qualitative agreement with the moisture content of green elm and linden leaves.

## Figures and Tables

**Figure 1 sensors-21-07694-f001:**
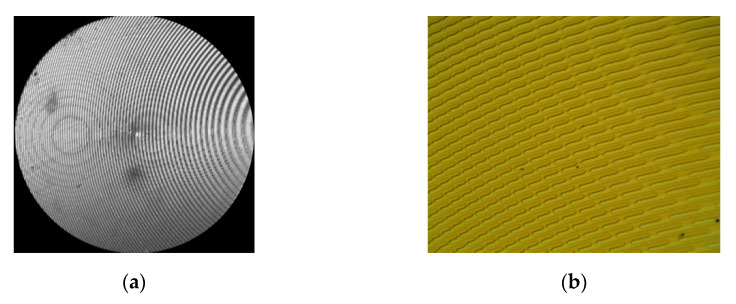
An optical microscope image of (**a**) an SDL (455 nm, 750 nm), (**b**) an SDL fragment with fine structure (455 nm, 750 nm), and (**c**) a profilogram of the SDL (455 nm, 750 nm) measured relative to the lens symmetry axis using a Tencor profiler.

**Figure 2 sensors-21-07694-f002:**
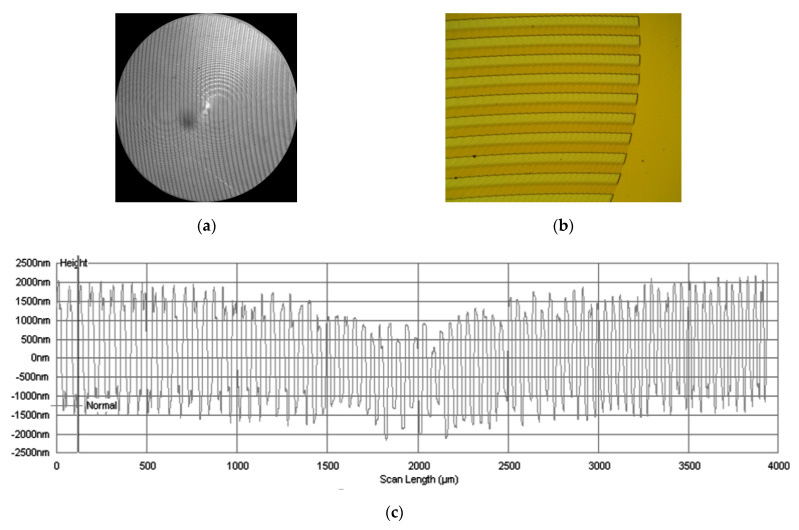
An optical microscope image of (**a**) an SDL (900 nm, 970 nm), (**b**) an SDL fragment with fine structure (900 nm, 970 nm), and (**c**) a profilogram of the SDL measured along the lens (900 nm, 970 nm) symmetry axis using a Tencor profiler.

**Figure 3 sensors-21-07694-f003:**
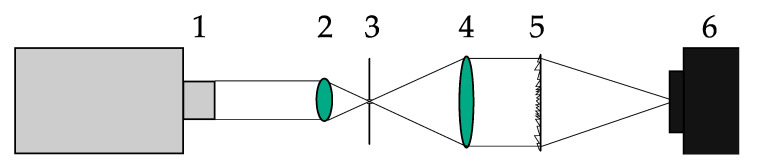
An experimental optical scheme for measuring the point spread function of the SDL: 1—a tunable laser NT-242, 2—a micro-objective, 3—a 10-µm pinhole, 4—a collimating lens, 5—an SDL, 6—a recording camera Basler acA 1920-40um (Basler, Ahrensburg, Germany).

**Figure 4 sensors-21-07694-f004:**
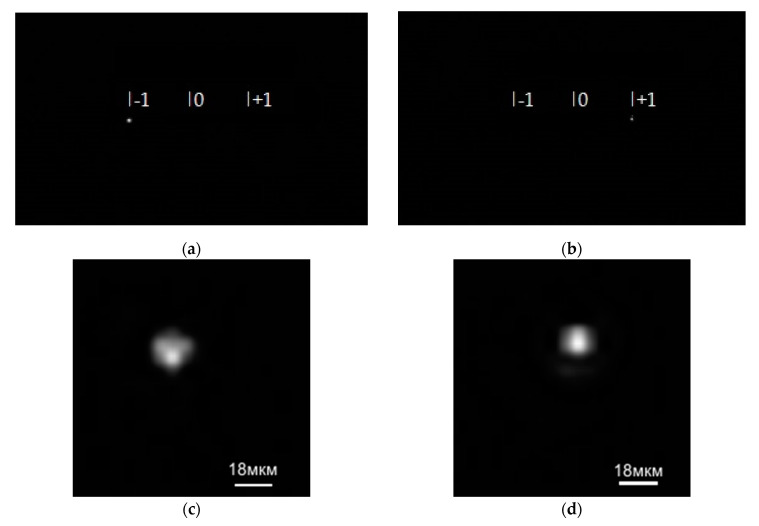
Diffraction orders on the imaging matrix from the SDL at wavelengths of (**a**) 455 nm, (**b**) 750 nm, and (**c**,**d**) their respective magnified fragments.

**Figure 5 sensors-21-07694-f005:**
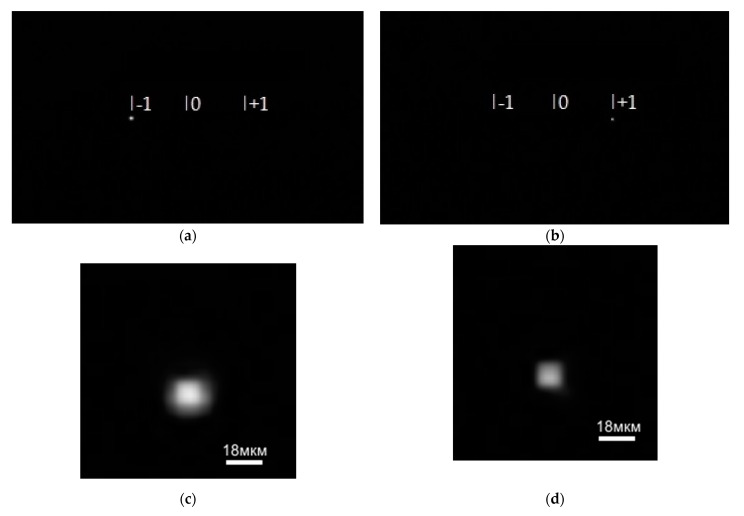
Diffraction orders on the matrix for wavelengths of (**a**) 900 nm, (**b**) 970 nm, and (**c**,**d**) their respective magnified images.

**Figure 6 sensors-21-07694-f006:**
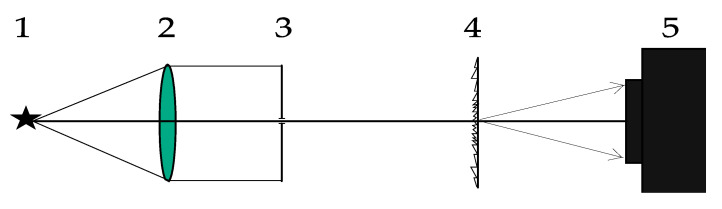
An experimental optical scheme: 1—an object, 2—a collimating lens, 3—an opaque screen with a pinhole, 4—an SDL, and 5—a camera.

**Figure 7 sensors-21-07694-f007:**
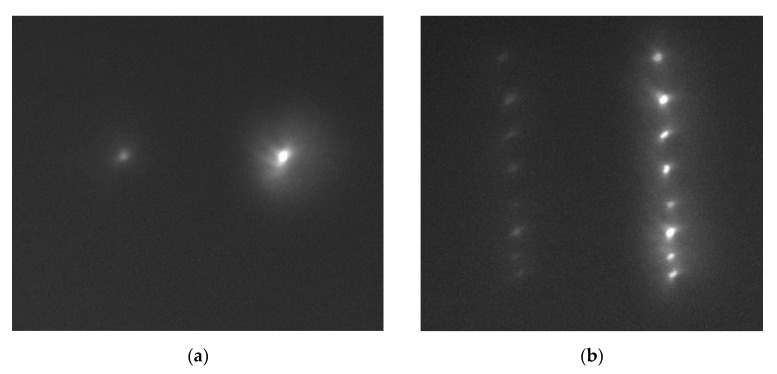
Matrix-displayed diffraction orders for wavelengths of 455 nm and 750 nm for (**a**) a single point source and (**b**) a line-arranged group of point sources.

**Figure 8 sensors-21-07694-f008:**
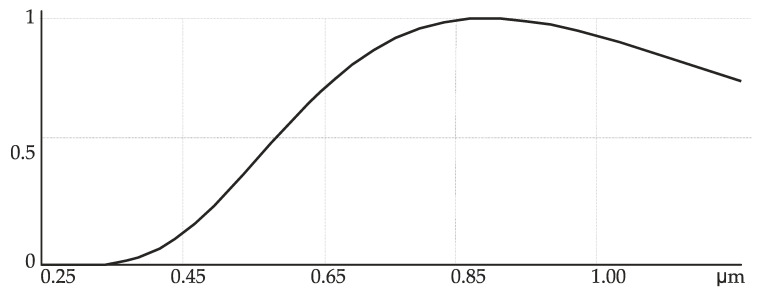
The curve of the relative spectral intensity of a halogen lamp.

**Figure 9 sensors-21-07694-f009:**
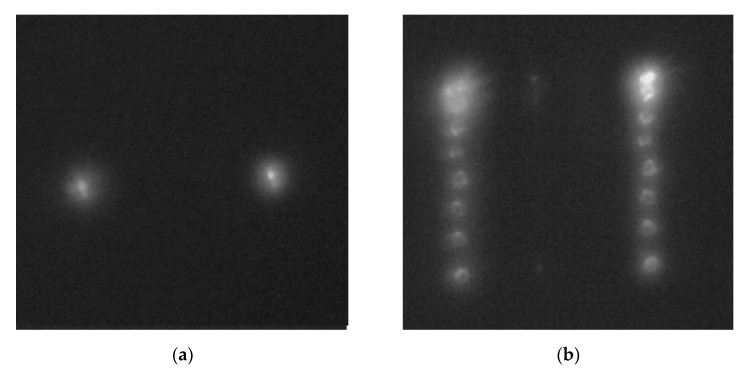
SDL-aided diffraction orders imaged on the matrix for wavelengths of 900 nm and 970 nm from (**a**) a single source and (**b**) a group of linearly arranged ~1-mm sources.

**Figure 10 sensors-21-07694-f010:**
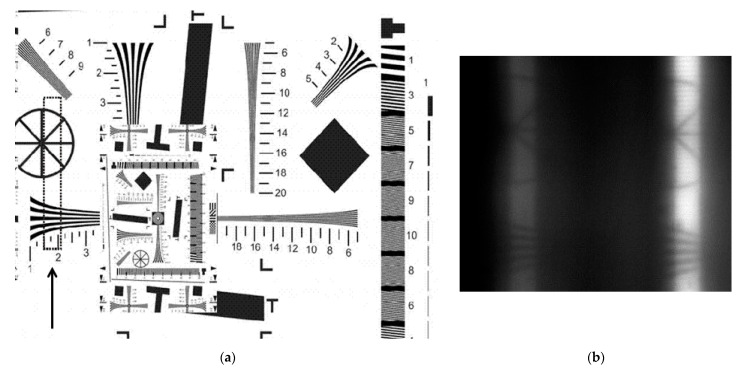
(**a**) An optical test pattern (the area covered is within a dashed-line rectangle and marked with an arrow) and (**b**) a fragment of the test pattern image obtained through a 315-um slit diaphragm, with the covered area marked by dashed lines.

**Figure 11 sensors-21-07694-f011:**
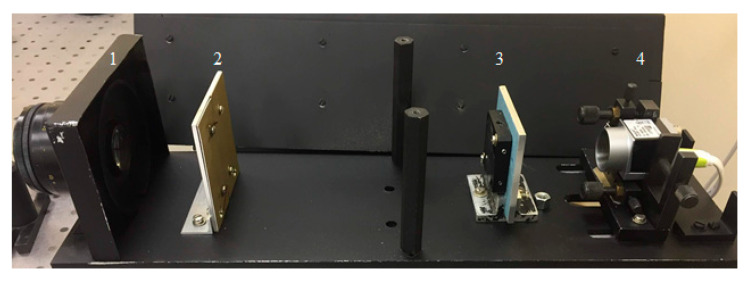
An experimental optical setup.

**Figure 12 sensors-21-07694-f012:**
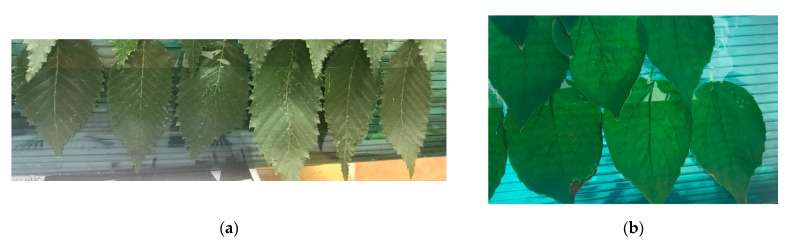
Leaves of (**a**) elm and (**b**) linden were used in the experiment.

**Figure 13 sensors-21-07694-f013:**
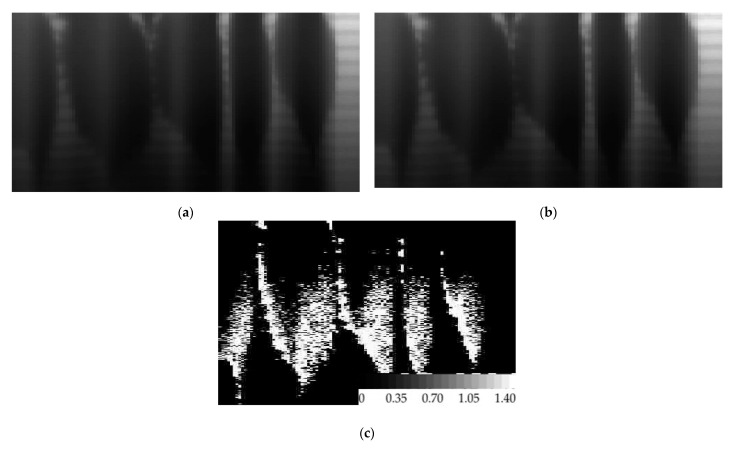
SDL-aided images of elm leaves at wavelengths of (**a**) 900 nm, (**b**) 970 nm, and (**c**) a water band index image of elm leaves.

**Figure 14 sensors-21-07694-f014:**
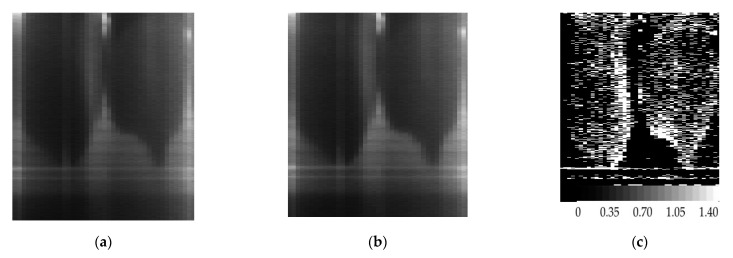
SDL-aided images of linden leaves at wavelengths of (**a**) 900 nm, (**b**) 970 nm, and (**c**) a water band index image of linden leaves.

## Data Availability

Not applicable.

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
