# Peer review of "Spectral Diffractive Lenses for Measuring a Modified Red Edge Simple Ratio Index and a Water Band Index"

_sensors, 2021, doi:10.3390/s21227694_

Round 1

Reviewer 1 Report

This is a systematical study of the use of SDLs in recording of radiance on wanted wavebands. SDLs were tested first using monocromatic light, then with a broadband light source and finally recording selected wavebands from plant leaves.

Two different indices relevant in vegetation monitoring were chosen as the targets for the SDLs.

Abstract says that both studied indices were tested with leaves. Please check if this was correct. I understood that the water index was tested with plant leaves.

How were the lower water contents of the heated parts estimated in addition to the index? Maybe feeling out or visually observing the lower hydrostatic pressure?

Consider adding scalebars in the microscopy images.

In the referee version at least, numbers and letters in the Tencor profile images are with too low resolution to be readable.

Especially in Fig. 4 b the zero diffraction order does nor seem to be in the center of the image - consider clarifying.

I do not see the dashed lines in Fig. 10 a.

I wonder, could you add a description of what part is mobile when scanning of the leaves was done in Fig. 11. Or did you move the leaves?

Fig. 13 c and Fig. 14 c, a scale bar for the index should be added.

The Introduction of this study presents a clear need for a lightweight, optimized systems for vegetation monitoring. Consider adding a description in the Discussion or Conclusios on what could be the next steps to develop the SDL system further to meet this need.

Author Response

The authors wish to thank the reviewers for their valuable and useful comments. We hope that the revision of the manuscript made to address the comments has improved the manuscript.

  1. Abstract says that both studied indices were tested with leaves. Please check if this was correct. I understood that the water index was tested with plant leaves.

 We have to admit that there was a mistake in the Abstract, as in fact only the water band index was experimentally built, and it stands to reason that the Red Edge Simple Ratio Index coincides with green leaves.

  1. How were the lower water contents of the heated parts estimated in addition to the index? Maybe feeling out or visually observing the lower hydrostatic pressure?

 For the moisture content of leaves to be objectively estimated, we utilized a conductometry technique, measuring the electrical conductivity of leaves. Relevant changes have been made in the text.

  1. Consider adding scalebars in the microscopy images.

 The scale bars have been added.

  1. In the referee version at least, numbers and letters in the Tencor profile images are with too low resolution to be readable.

 In the profilograms, the informative fragments have been highlighted while the housekeeping data has been dropped. For better visibility, the illustrations have been inverted.

  1. Especially in Fig. 4 b the zero diffraction order does nor seem to be in the center of the image - consider clarifying.

 It seems rather hard to center an experimental image on the zero order, but in the revised version in Figs. 4a,b and 5a,b we have done so.

  1. I do not see the dashed lines in Fig. 10 a.

 In Fig. 10a, we magnified the image leaving just the central part of the test pattern and marking it with an arrow.

  1. I wonder, could you add a description of what part is mobile when scanning of the leaves was done in Fig. 11. Or did you move the leaves?

 In our experiment, it was leaves that were moved on a special moving table. The relevant description has been added.

  1. Fig. 13 c and Fig. 14 c, a scale bar for the index should be added.

 A scale of index values has been added.

  1. The Introduction of this study presents a clear need for a lightweight, optimized systems for vegetation monitoring. Consider adding a description in the Discussion or Conclusios on what could be the next steps to develop the SDL system further to meet this need.

Description has been added.

Reviewer 2 Report

   This work proposes the use a Spectral Diffractive Lens (SDL) to spatially split the diffractive orders of normally incident light, as a means to reduce the cost of hyperspectral imagers while achieving similarly NDVI-like results. Specifically, their focus is to apply this technology to agriculture; the laboratory experiment focuses on using elm and linden leaves. The motivation for the work is to reduce the cost of multispectral imaging.

   The premise for the work is strong, and the idea is intriguing. However, the execution of the experiment is lacking, and other resources have not been considered. For instance, a recent paper from the Journal of Chemical Education (https://doi.org/10.1021/acs.jchemed.0c00407) demonstrates a low-cost multispectral imager, with a very similar goal. Their study provides three examples that can be easily confirmed by the human eye (ex. ripened vine tomatoes). To confirm their multispectral imaging results, a human can look at the tomato and say, "yes, this is ripe, and this one is not. These results match our spectral imager results."

   One problem with the submitted paper is that the results cannot be easily confirmed by eye -- for instance, the authors have not objectively demonstrated that the water content (Figs. 13c, 14c) is allocated in the way that the multispectral imaging suggests.  Has polarization been taken into account? Why haven't the green healthy leaves been directly compared to brown dying leaves of the same species? Additionally, the use of diffractive optics in multispectral imaging has been demonstrated and previously published as well (https://doi.org/10.1117/12.2557106). 

   Given the lack of objective evidence in the experiment results and the presence of similar technology in the literature, the unique impact of this technology remains unclear.

Author Response

The authors wish to thank the reviewers for their valuable and useful comments. We hope that the revision of the manuscript made to address the comments has improved the manuscript

  1. The premise for the work is strong, and the idea is intriguing. However, the execution of the experiment is lacking, and other resources have not been considered. For instance, a recent paper from the Journal of Chemical Education (https://doi.org/10.1021/acs.jchemed.0c00407) demonstrates a low-cost multispectral imager, with a very similar goal. Their study provides three examples that can be easily confirmed by the human eye (ex. ripened vine tomatoes). To confirm their multispectral imaging results, a human can look at the tomato and say, "yes, this is ripe, and this one is not. These results match our spectral imager results."

To address this comment, we briefly discussed the paper mentioned by the Reviewer. It is worth mentioning, although the method proposed there is exclusively for the laboratory use.

  1. One problem with the submitted paper is that the results cannot be easily confirmed by eye -- for instance, the authors have not objectively demonstrated that the water content (Figs. 13c, 14c) is allocated in the way that the multispectral imaging suggests.  Has polarization been taken into account? Why haven't the green healthy leaves been directly compared to brown dying leaves of the same species? Additionally, the use of diffractive optics in multispectral imaging has been demonstrated and previously published as well (https://doi.org/10.1117/12.2557106). 

We deemed it very important to identify dry areas of leaves visually indiscernible from the wet ones using a water band index. Otherwise, the use of the water band index does not make much sense because it is easier to utilize a conventional RGB image. The leaf dryness was estimated using a conductometry technique (with its description added to the text). Although this technique is not very accurate in the case of an individual leaf, it still enables dry leaf parts to be discerned from the wet ones.

Polarization of light was not accounted for.

Brown dying leaves are almost invisible in the water band index image (see black noisy image; the experiment was additionally conducted to address the comments made).

We added to the Introduction the proposed paper which really offers an alternative approach to that proposed in our work.  

Round 2

Reviewer 2 Report

All concerns were nicely addressed by the authors.